# A Zebrafish Model for a Rare Genetic Disease Reveals a Conserved Role for FBXL3 in the Circadian Clock System

**DOI:** 10.3390/ijms23042373

**Published:** 2022-02-21

**Authors:** Shir Confino, Talya Dor, Adi Tovin, Yair Wexler, Zohar Ben-Moshe Livne, Michaela Kolker, Odelia Pisanty, Sohyun Kathy Park, Nathalie Geyer, Joel Reiter, Shimon Edvardson, Hagar Mor-Shaked, Orly Elpeleg, Daniela Vallone, Lior Appelbaum, Nicholas S. Foulkes, Yoav Gothilf

**Affiliations:** 1School of Neurobiology, Biochemistry and Biophysics, Faculty of Life Sciences, Tel-Aviv University, Tel-Aviv 6997801, Israel; shirconfino@gmail.com (S.C.); yairwex@gmail.com (Y.W.); zohar.bm@gmail.com (Z.B.-M.L.); odeliap@gmail.com (O.P.); 2Pediatric Neurology Unit, Hadassah Medical Center and Faculty of Medicine, Hebrew University of Jerusalem, Jerusalem 91120, Israel; talyad@hadassah.org.il (T.D.); simon@hadassah.org.il (S.E.); 3The Faculty of Life Sciences and the Multidisciplinary Brain Research Center, Bar-Ilan University, Ramat-Gan 5290002, Israel; aditovin@gmail.com (A.T.); lior.appelbaum@biu.ac.il (L.A.); 4School of Zoology, Faculty of Life Sciences, Tel-Aviv University, Tel-Aviv 6997801, Israel; kolkermi@mail.tau.ac.il; 5Institute of Biological and Chemical Systems, Karlsruhe Institute of Technology, 76344 Eggenstein-Leopoldshafen, Germany; kathy_p927@hotmail.com (S.K.P.); nathalie.geyer@kit.edu (N.G.); daniela.vallone@kit.edu (D.V.); nicholas.foulkes@kit.edu (N.S.F.); 6Pediatric Pulmonary & Sleep Unit, Hadassah Medical Center and Faculty of Medicine, Hebrew University of Jerusalem, Jerusalem 91120, Israel; rjoely@hadassah.org.il; 7Department of Genetics, Hadassah Medical Center and Faculty of Medicine, Hebrew University of Jerusalem, Jerusalem 91120, Israel; hagarmor@hadassah.org.il (H.M.-S.); elpeleg@hadassah.org.il (O.E.); 8Sagol School of Neuroscience, Tel-Aviv University, Tel-Aviv 6997801, Israel

**Keywords:** FBXL3, zebrafish, circadian clock, rare genetic disease

## Abstract

The circadian clock, which drives a wide range of bodily rhythms in synchrony with the day–night cycle, is based on a molecular oscillator that ticks with a period of approximately 24 h. Timed proteasomal degradation of clock components is central to the fine-tuning of the oscillator’s period. FBXL3 is a protein that functions as a substrate-recognition factor in the E3 ubiquitin ligase complex, and was originally shown in mice to mediate degradation of CRY proteins and thus contribute to the mammalian circadian clock mechanism. By exome sequencing, we have identified a *FBXL3* mutation in patients with syndromic developmental delay accompanied by morphological abnormalities and intellectual disability, albeit with a normal sleep pattern. We have investigated the function of FBXL3 in the zebrafish, an excellent model to study both vertebrate development and circadian clock function and, like humans, a diurnal species. Loss of *fbxl3a* function in zebrafish led to disruption of circadian rhythms of promoter activity and mRNA expression as well as locomotor activity and sleep–wake cycles. However, unlike humans, no morphological effects were evident. These findings point to an evolutionary conserved role for FBXL3 in the circadian clock system across vertebrates and to the acquisition of developmental roles in humans.

## 1. Introduction

An intrinsic timing mechanism in animals, the circadian clock, drives a variety of physiological and behavioral daily rhythms in synchrony with the environmental day-night cycle. Central to this system is an evolutionary conserved, self-sustaining molecular oscillator that generates circadian (approximately 24 h) rhythms of gene expression and other cellular functions [1]. The underlying mechanism of this molecular oscillator is based on positive and negative clock factors organized in the context of transcriptional regulatory feedback loops [1,2]. The positive elements, the brain and muscle ARNT (aryl hydrocarbon receptor nuclear translocator)- like (BMAL) protein and the circadian locomotor output cycles kaput (CLOCK) protein, which belong to the basic helix–loop–helix transcription factor family drive the expression of clock-controlled genes by binding to E-box enhancer regulatory regions (5′-CACGTG-3′) within their promoters [3,4]. They also drive the expression of the negative clock factors, the period (PER) and cryptochrome (CRY) proteins, which translocate into the nucleus as heterodimers and then directly interact with the BMAL-CLOCK heterodimer and inhibit its function, thereby down-regulating their own expression [4,5]. When PER and CRY levels drop, gene activation is resumed and the molecular cycle, which takes about 24 h to complete, starts again. An important process that determines the pace of this molecular oscillator is the degradation of the negative clock components. The importance of CRY degradation was demonstrated by the characterization of FBXL3-deficient mice that exhibit a free-running rhythm of 27 h [6]. The F-box and leucine rich repeat protein 3 (FBXL3) functions in the Skp1-Cullin1-Fbox (SCF) E3 ubiquitin ligase complex to mediate proteasomal degradation of CRY proteins, consequently affecting the pace of the molecular oscillations and subsequent behavioral rhythms [6,7,8]. Thus, the *Fbxl3* mutant mice exhibit a behavioral phase-delay phenotype and a phase-delay of membrane excitability in neurons of the ventral suprachiasmatic nucleus (SCN) [9], the site of the master clock in mammals.

Very few examples of *FBXL3* mutations have been encountered in humans [10]. We identified three patients who are homozygous for a *FBXL3* loss-of-function (LOF) mutation. These patients exhibit prenatal macrocephaly, midface hypoplasia, short stature, hypotonia and intellectual disability, similar to a recently reported case of three unrelated families bearing biallelic LOF variants of *FBXL3* [10]. Given the reported contribution of FBXL3 function to the molecular clock in mice and in human cell lines, we also monitored their sleep pattern but found no evidence for any sleep disturbances. Therefore, in humans, the most obvious phenotypes of the *FBXL3* mutation are developmental delay and morphological abnormalities, while in the mouse model only disrupted circadian rhythms have been reported [6,8]. However, in both mice and human cell lines, the role of FBXL3 is to regulate the pace of CRY turnover and of the molecular oscillator [7,8,11].

In order to scrutinize the effect of *FBXL3* LOF on both morphogenesis and the circadian clock in more detail, we next chose to investigate the role of FBXL3 in zebrafish. In addition to being a powerful model for studying developmental genetics and morphological abnormalities [12,13,14], the zebrafish is a diurnal species that has been widely used as a model for circadian clock research [15,16,17].

## 2. Results

### 2.1. Clinical Evaluation and Exome Analysis

The index-patient (IV-1, Figure 1A), the first child to first cousin parents, was examined at 13 years of age. He exhibited macrocephaly (+3SD), short stature (-4SD), hypotonia and moderate intellectual disability. His younger sister (patient IV-3, Figure 1A) had similar manifestations with prenatal macrocephaly, muscle hypotonia since birth and delayed psychomotor development. Their younger brother, the fifth child in the family (patient IV-5, Figure 1A), was noted to have macrocephaly and midfacial hypoplasia as early as 13 weeks of gestation. Both parents (III-5 and III-6, Figure 1A) and two other siblings (IV-2 and IV-4, Figure 1A) were healthy. Clinical descriptions are provided in Figure 1.

With the goal of understanding the etiology of this rare neurodevelopmental syndrome, and to enable future identification of carriers, we chose to study the genetic basis of this disease. Trio exome analysis of the index-patient (IV-1) yielded 61.0 million mapped reads with a mean coverage of 81X. Following alignment to the reference genome (hg19) and variant calling, we performed a series of filtering steps under the hypothesis of a recessively inherited, rare allele. These included removing variants that were called less than X8, were off-target, heterozygous, synonymous, X chromosome variants, had MAF  >  0.1% at gnomAD v2.1.1. (genome aggregation database, URL: https://gnomad.broadinstitute.org/ (accessed on 1 April 2019)) or MAF  >  4% at the Hadassah in-house database. Of the six remaining variants (Appendix A), only one segregated with the disease in the family. This was hg19 Chr13:g.77581816G>A, NM_012158.4:c.751C>T, p.Arg251Ter (also hg38 chr13:77007681G>A) in the *FBXL3* gene (Figure 1B), predicted to cause a homozygous LOF. Importantly, our work, combined with the previous report [10], provides valuable confirmation that *FBXL3* LOF mutations in humans cause developmental phenotypes including macrocephaly, short stature, hypotonia and delayed psychomotor development.

### 2.2. Sleep Assessment of Patients

Given the documented contribution of FBXL3 to circadian clock function and thereby activity rhythms in mice, we chose to explore the sleep–wake cycles of these patients. Total sleep time for the major daily sleep period, sleep latency (time to fall asleep), sleep efficiency (sleep duration out of time in bed), time awake after sleep onset and general sleep patterns were assessed for two of the affected patients using a wrist-based activity monitor for a period of 14 days (Figure 2). Both patients exhibited regular sleep patterns. Patient IV-3, 18 years old at the time of sleep assessment, slept 8:29 h on average, between 22:14 and 7:44, with a sleep efficiency of 89.5%. Patient IV-5, 10 years old upon sleep assessment, slept 8:41 h on average, between 23:41 and 8:53, with a sleep efficiency of 92.3%, suggestive of a mildly delayed sleep phase. Neither patient showed significant daytime sleep. At the time of assessment, the patients were routinely attending special education framework, whereas their healthy siblings were on school leave and home lockdown (secondary to the corona-virus outbreak) and consequently were reported to exhibit poor sleep hygiene.

### 2.3. Establishing a FBXL3-Deficient Zebrafish Line

Zebrafish *fbxl3a* (RefSeq NM_001005773.3) encodes a 431 amino acid protein which is homologous to the human FBXL3 protein with 86% identity. It is located on chromosome 9, and, as in humans and mice, it contains four translated exons following a 5’UTR exon (Figure 3A), and has conserved synteny with flanking genes (Figure 3B). Using CRISPR/Cas9 technology, a stable *fbxl3a* LOF mutant line was established that has a single base pair (bp) deletion in exon 2 which introduces a frame shift and a premature stop codon, and is predicted to result in a truncated protein of 94 amino acids of which only the first 14 amino acids are present in the wildtype (WT) protein (Figure 3A and Appendix A). Subsequent experiments were performed on the progeny of intercrosses between heterozygous (*fbxl3a*^+/−^) fish, resulting in a Mendelian distribution of WT, heterozygous and homozygous individuals. Other experiments were performed on the progeny of intercrossed mutant fish (*fbxl3a*^−/−^) along with progeny of intercrossed WT sibling fish (*fbxl3a*^+/+^) as controls.

### 2.4. FBXL3a Deficiency Does Not Affect Zebrafish Morphology

Contrary to the previous observations in mice, the primary phenotype of FBXL3 LOF in humans appear to consist of morphological abnormalities. To test the effect of FBXL3a deficiency on zebrafish morphology, we conducted morphological measurements of WT siblings, heterozygous and homozygous *fbxl3a* mutant fish. Progeny of a cross between heterozygous (*fbxl3a*^+/−^) fish were inspected at the beginning of the pharyngula period (24 h post-fertilization, hpf) and its end (48 hpf), when spontaneous embryonic movements develop into behavioral responses to touch, heartbeat initiates and circulation appears. Morphological traits were inspected again at the juvenile stage (10 weeks of age) by measuring several cranial and body bones (Appendix A). At both the embryonic stages and the juvenile stage, no evidence for structural abnormalities was detected (Appendix A).

### 2.5. FBXL3a Deficiency Affects Circadian Rhythms of Locomotor Activity in Zebrafish

We next assayed rhythmic locomotor activity as a measure of circadian clock regulation at the whole animal level. The locomotor activity of clock-entrained larvae was monitored under constant dim light (DimDim, 7.5 lux) for three days (6–8 days post-fertilization, dpf). Under these constant conditions, WT larvae exhibit robust circadian rhythms of locomotor activity [18,19,20]. The average locomotor activity of WT and *fbxl3a*^−/−^ larvae is plotted in Figure 4A, revealing a 7.2 h phase-delay of the *fbxl3a*^−/−^ larvae activity compared with their WT siblings (Figure 4B). Furthermore, a significantly lower median G-factor score [18] was obtained for the *fbxl3a*^−/−^ larvae activity compared with their WT siblings, indicating greater divergence from 24 h rhythmicity (Figure 4C). Indeed, *fbxl3a*^−/−^ larvae exhibit highly variable clock-controlled period lengths of activity rhythms compared with their WT siblings. Clustering analysis based on individual data separates WT and mutant fish into two main clusters (*p* < 0.001); the resulting heat map (Figure 4D) emphasizes the disordered locomotor activity rhythms of *fbxl3a*^−/−^ larvae, as compared to their WT siblings. Interestingly, heterozygous (*fbxl3a*^+/−^) larvae display a mild intermediate phenotype of approximately 3 h phase-delay compared with WT larvae, and a lower G-factor score (Figure 4F–H). To test whether *fbxl3a* deficiency might simply impair larval motility, larvae were exposed to dark flashes of 10 s each during the light phase, a paradigm known to induce immediate bursts of activity [21]. No statistical difference was observed between the dark flash-induced activity of WT and *fbxl3a* mutant larvae (Figure 4I), thus ensuring that the mutation did not impair overall larval mobility. These results indicate that *fbxl3a* deficiency affects circadian rhythms of locomotor activity in zebrafish. Under light/dark (LD) cycles, *fbxl3a* mutant larvae exhibit daily rhythms, albeit with relatively blunted locomotor activity during the day (light phase, 770 lux) and slightly higher activity during the night (dark phase) compared with their WT siblings. Interestingly, the *fbxl3a* mutants produce an acute behavioral response to the light-to-dark transition but lack a response to the dark-to-light transition (Figure 4J). This may be the result of the delayed circadian phase in which the mutants are upon lights-on.

In adult zebrafish, spawning occurs in the morning within about an hour following lights-on. Since we noticed a delay in spawning time, we monitored and recorded the time of spawning of *fbxl3a* mutant fish and their WT siblings throughout the study. A markedly delayed spawning time was observed for the adult mutant fish (Figure 4E), as predicted from their delayed circadian phase. This reflects the physiological importance of *fbxl3a* function also under regular LD conditions.

### 2.6. FBXL3a Deficiency Affects Sleep–Wake Cycles in Zebrafish

FBXL3 deficiency disrupts rhythms of locomotor activity in mice [6,8] and zebrafish larvae (Figure 4). However, in humans, despite its effect on the molecular oscillator in cell lines, FBXL3 deficiency did not affect the night-time sleep pattern of the examined patients (Figure 2). To determine whether *fbxl3a* LOF affects sleep–wake cycles in zebrafish, we analyzed sleep patterns under different lighting conditions. Sleep in zebrafish larvae has been defined as an inactive bout of more than one minute, based on the finding that larvae exhibit reduced responsiveness to stimuli after one minute of inactivity [22]. Using this criterion, sleep time was determined under LD and DimDim conditions. The analysis revealed an abnormal sleep pattern of *fbxl3a*^−/−^ larvae under LD, with increased sleep time during the day (light phase, Figure 5A). The average sleep time during day and night under LD conditions (Figure 5B) emphasizes the day/night sleep pattern differences between WT and *fbxl3a* mutant larvae. Moreover, under DimDim conditions, an arrhythmic sleep pattern of *fbxl3a*^−/−^ larvae emerges (Figure 5C). Interestingly, sleep is also disrupted in heterozygous (*fbxl3a^+/^*^−^) larvae under DimDim conditions (Figure 5D).

### 2.7. FBXL3a Deficiency Affects the Rhythmic Expression of Pineal aanat2

The pineal gland of zebrafish, as well as other non-mammalian vertebrates, is considered a key element in the circadian clock system. It is a classical photoreceptive organ and contains an intrinsic circadian oscillator that drives rhythms of melatonin production and secretion. By these rhythms, the pineal gland transduces photoperiodic information into a hormonal signal, which in turn influences many daily rhythms including the sleep–wake cycle and seasonal changes [23]. Particularly, in zebrafish, exogenous melatonin has been shown to induce sleep [24,25,26,27]. An extensively studied gene exhibiting pineal-specific expression pattern and circadian rhythms of expression is arylalkylamine-*n*-acetyltransferase-2 (*aanat2*), which encodes the rate-determining enzyme in the melatonin synthesis pathway. Importantly, zebrafish *aanat2* is an E-box regulated clock-controlled gene; accordingly, it exhibits rhythmic expression that is driven by the molecular oscillator [20]. We therefore hypothesized that *aanat2* expression would be affected by *fbxl3a* LOF.

*Fbxl3a* mutant larvae and WT siblings’ progeny were entrained for 3 days under a LD regime, after which they were sampled at 4 h intervals throughout a 24 h period under LD or constant dark (DD) conditions. Pineal *aanat2* mRNA levels were semi-quantified by whole mount in situ hybridization (ISH) followed by optical density measurements of the signal. Consistent with previous reports [20,28,29,30], the levels of *aanat2* mRNA in the pineal glands of WT larvae were low during the day (or subjective day) and increased throughout the night, reaching a peak towards the middle to end of the night period. On the other hand, under both LD and DD conditions, irregular levels of pineal gland *aanat2* mRNA were detected in the *fbxl3a*-deficient larvae, characterized by significantly higher overall expression and a delayed phase with respect to their WT siblings (Figure 6). These results suggest that *fbxl3a* deficiency also affects the molecular oscillator within the pineal gland, a key organ in the circadian clock system of fish.

### 2.8. FBXL3a Deficiency Disrupts Molecular Circadian Oscillations

In order to investigate the effect of *fbxl3a* LOF on the molecular oscillator, cell lines were established from 24 hpf *fbxl3a* mutant and progeny of WT sibling embryos. These cell lines were transiently transfected with the *per1b:luc* promoter–reporter construct and rhythmicity was assessed by a bioluminescence assay [31]. Consistent with the in vivo assays, the results indicate impaired rhythmic *per1b* promoter activity, characterized by a phase-delayed rhythm and period lengthening under DD conditions in the *fbxl3a*^−/−^ cells with respect to the WT control cell line (Figure 7A–D). Furthermore, to test whether the phase-delay of bioluminescence rhythms driven by the core molecular oscillator is due to a direct effect of FBXL3a deficiency, we exogenously expressed the zebrafish WT FBXL3a protein in the *fbxl3a*-deficient cells, together with the *per1b:luc* promoter–reporter construct. Importantly, the expression of the WT form of FBXL3a (Appendix A) was able to partially rescue the bioluminescence phase-delay and period lengthening in the *fbxl3a*-deficient cells (Figure 7A–D), indicating that *fbxl3a* LOF directly induced the phase-delay.

The rhythmic expression of *per1b* is dependent on E-box-mediated rhythmic transcriptional activation by the BMAL-CLOCK heterodimer [15]. Therefore, to further assess the effect of *fbxl3a* deficiency on the core molecular oscillator, the cell lines were transfected with an *E-box:luc* construct, in which the luciferase reporter gene is driven by 4 tandem repeats of the canonical E-box enhancer element, cloned upstream of a minimal promoter [31]. Bioluminescence was measured during exposure to five LD cycles followed by 5 days under DD (Figure 7E). While the WT cell line exhibit a robust oscillation under LD with a precise 24 h period that dampens under DD, mutant cells exhibit a low amplitude oscillation with a phase-delay of approximately 4 h, accompanied by high variability in period length (Figure 7F,G), followed by a near loss of oscillation under DD.

Taken together, these in vitro data suggest a major role for *fbxl3a* in the normal function of the core molecular oscillator in zebrafish.

## 3. Discussion

In the present study, we have identified a *FBXL3* LOF mutation in a consanguineous family in which homozygous patients were affected with developmental delay, morphological abnormalities and moderate intellectual disability. FBXL3 was shown to serve as a key circadian clock component in mice, but no evidence for morphological defects was reported in that model. Therefore, we chose to investigate the role of FBXL3 in zebrafish, a diurnal vertebrate which has been widely used for studying the circadian clock and is ideally suited to study the mechanisms underlying morphogenesis.

The zebrafish molecular oscillator was disrupted by the *fbxl3a* LOF mutation, as indicated by the dynamic expression pattern of the *per1b* promoter or synthetic E-box promoter in cell lines derived from the mutant fish. Importantly, this disruption was partially rescued by ectopic expression of WT FBXL3a protein. The relatively low efficiency of rescue is most likely the consequence of the transient transfection of the cells with the FBXL3a expression construct which typically results in heterogeneity of expression levels within individual transfected cells. Nevertheless, the effect of the *fbxl3a* mutation on the molecular oscillator in zebrafish cells is consistent with similar affects in human- and mouse-derived cell lines and points to a highly conserved contribution of FBXL3 to clock function in vertebrates.

Clock-controlled rhythms of locomotor activity and sleep–wake cycles were also disrupted in *fbxl3a* mutant fish. Moreover, rhythmic expression of pineal gland *aanat2*, a clock-controlled E-box-regulated gene which encodes the rate-determining enzyme in the melatonin synthesis pathway, was disrupted. Since melatonin is known to promote sleep in zebrafish [24,25,26,27], it is tempting to speculate that the effect of FBXL3-deficiency on sleep is partly related to alteration in melatonin secretion.

In contrast to our findings in zebrafish, the *FBXL3* mutation in the human patients did not affect their sleep–wake cycle. This was unexpected because mutations in CRY protein, the targets of FBXL3, are associated with Familial Advanced Sleep Phase [11] and Familial Delayed Sleep Phase [32] syndromes, and because pharmacological inhibition of the FBXL3-CRY interaction has been shown to affect the period of the molecular oscillator in human cells [33]. The resilience of the patients’ sleep–wake cycle to the *FBXL3* mutation may be explained by a rigid daily routine of attending a special education framework and by the associated day/night cycle in their environment. Indeed, in the diurnal zebrafish, the sleep–wake pattern was only mildly affected under LD conditions, possibly reflecting a masking effect of light and dark, suggesting that under natural light conditions and a daily routine, the affected patients can also maintain normal sleep–wake patterns. Thus, although one *FBXL3*-variant patient previously self-reported sleep disturbances [10], additional cases should be tested under defined protocols before concluding about the contribution of *FBXL3* LOF to the sleep–wake cycle, as well as other circadian rhythms and their synchronization in humans. Moreover, given that abnormal rhythmic locomotor activity and sleep patterns were also observed in heterozygous (*fbxl3a*^+/−^) fish, the penetrance of the mutation in the relevant human population and the possibility that heterozygous carriers and their families exhibit certain circadian clock-related deficiencies warrants further investigation under controlled conditions.

The presented molecular and behavioral analyses in zebrafish, together with the demonstrated effect of *FBXL3* mutation in mice and in human cell lines, emphasize the conserved role of FBXL3 in maintaining circadian oscillations throughout the vertebrates, but are inconsistent with the morphological and neurodevelopmental affects observed in the human patients. It would be attractive to speculate that the contribution of FBXL3 deficiency to these apparent developmental defects in humans is related to its effect on the expression of clock genes. Nevertheless, given that in both zebrafish and mice, FBXL3 deficiency affects the circadian clock but does not result in morphological changes, it seems likely that FBXL3 has acquired distinct developmental roles in humans, independently of the core clock mechanism. Another explanation for this discrepancy is that the *FBXL3* mutations in humans (this study and Ansar et al. [10]) exert a cis effect on the expression of an adjacent gene, *CLN5*. Recessive mutations in *CLN5* are associated with neuronal ceroid lipofuscinosis-5, a syndrome characterized by ataxia, seizures, myoclonus, visual impairment and cognitive/motor decline. In humans, as opposed to mice and zebrafish, *CLN5* and *FBXL3* are adjacent to each other (Figure 3B), and the described *FBXL3* mutations are located in proximity to *CLN5*, in the last exon of *FBXL3*. Although one *FBXL3* mutation with similar adverse effects is distantly located in exon 3 (Figure 3A), the possibility of cis effect warrants further investigation. Moreover, these two closely located genes were shown to form a fused transcript in acute myeloid leukemia cells [34]. Although the function of the *CLN5*-*FBXL3* fused transcript is as yet unknown, the effect of the human *FBXL3* mutations on its formation warrants further investigation, obviously requiring the recruitment of several cases.

Currently, a more likely explanation for the human-specific effects is that FBXL3 has additional roles in humans. Indeed, it has been shown that the FBXL3-CRY1/2 complexes can recruit and target many additional human proteins for proteasomal degradation [35], some of which may exert developmental defects upon mis regulation. For example, one of the proteins shown to bind the FBXL3-CRY1/2 complex is a developmentally regulated RNA-specific adenosine deaminase (ADAR). Dominant-negative mutations in the *ADAR* gene lead to Aicardi-Goutières syndrome, an inherited encephalopathy characterized by severe neurological dysfunctions and psychomotor retardation. Additionally, some of the FBXL3-CRY1/2 recruited proteins, such as c-MYC [36] and TLK2 [35], are involved in cell cycle regulation, providing another explanation for the observed circadian clock-gating of cell cycle, and justifying follow-up of the very few *FBXL3* LOF patients for cell cycle-related diseases such as cancers. Thus, it is therefore conceivable that FBXL3 has additional targets in humans; future identification of such targets for FBXL3-mediated ubiquitination in humans, in comparison to fish and mice, may reveal the molecular mechanisms underlying these observed evolutionary changes in FBXL3 function. The mutant zebrafish line and the derived cell lines have therefore provided a new tool for studying the function of FBXL3, and demonstrate the general utility of zebrafish as a model for studying circadian clock-related disorders.

In conclusion, the phenotype of FBXL3 LOF mutation in zebrafish reinforce the notion that in vertebrates, FBXL3 plays a highly conserved role in maintaining normal circadian clock oscillations. However, the morphological and neurodevelopmental affects observed in the human patients point to some degree of species-specific diversity in FBXL3 function that extends beyond the circadian clock.

## 4. Materials and Methods

### 4.1. Exome Analysis

Following informed consent, exome sequencing analysis was performed on DNA extracted from whole blood of the proband (individual IV-1) and the parents, as previously described [37,38,39]. Exonic sequences were enriched from the genomic DNA samples using SureSelect Human All Exon v.5 50 Mb Kit (Agilent Technologies, Santa Clara, CA, USA). Sequences were determined by HiSeq2500 sequencing system (Illumina, San Diego, CA, USA) as 100 bp paired-end runs. Data analysis including read alignment and variant calling was performed by DNAnexus software (Palo Alto, CA, USA) using the default parameters with the human genome assembly hg19 (GRCh37) as reference. Filtering was performed as described [37]. The trio exome analysis yielded 61.0 million mapped reads, with a mean coverage of 81X.

### 4.2. Segregation Analysis

An amplicon containing the *FBXL3* variant was amplified by conventional PCR of genomic DNA from proband and all available parents and siblings, and analyzed by Sanger dideoxy nucleotide sequencing.

### 4.3. Actigraphy and Patients Sleep Pattern Detection

Patients wore an actigraph (Actiwatch Spectrum Plus, Phillips Respironics, Andover, MA, USA) on their wrist continuously for 14 days. Data was then downloaded using the actiwatch software package. The analysis included the total sleep time for the major daily sleep period, sleep latency (time to fall asleep), sleep efficiency (sleep duration out of time in bed), time awake after sleep onset and general sleep patterns.

### 4.4. Establishment of the fbxl3a Mutant Zebrafish Line

The CRISPR-Cas9 system was used to establish the *fbxl3a*^−/−^ line, registered in the Zebrafish Model Organism Database (ZFIN) as *fbxl3a^tlv08^*. The Cas9 and sgRNA zebrafish-optimized expression vectors were obtained from Addgene (pMLM3613 and pDR274, Addgene plasmids #42251 and #42250, respectively). In order to prepare the sgRNA, two *fbxl3a*-specific oligos were designed to match the target sequence (5′-GGACAGCACCTCATCATGCG-3′) in exon 2. These oligos were denatured at 95 °C for 5 min, then gradually cooled down to room temperature and kept at 4 °C. Before cloning, annealing of the oligos was confirmed in 2% agarose gel. The annealed oligos were cloned into pDR274 between the *BsaI* restriction sites and transformed into bacteria. A bacterial clone was selected for further propagation based on plasmid sequencing. In order to synthesize the specific sgRNA, the DR274 plasmid containing the annealed oligos was linearized with *DraI* and cleaned using the standard phenol-chloroform procedure, followed by purification by PureLink PCR Purification Kit (#K31000, Life Technologies, Carlsbad, CA, USA). The sgRNA was synthesized using the T7 High Yield RNA Synthesis Kit (New England Biolabs, Hitchin, UK). In order to prepare Cas9 mRNA, pMLM3613 was linearized by *AgeI*, and mRNA was synthesized using the mMESSAGE mMACHINE T7 Kit (Life Technologies, Carlsbad, CA, USA).

One-cell stage WT zebrafish embryos were microinjected with mixture of Cas9 mRNA (300 ng/µL) and transcribed sgRNA (12.5 ng/µL). To test the efficiency of the CRISPR system, 10 one-dpf embryos were screened for *fbxl3a*-specific mutation using PCR amplification of genomic DNA (Forward primer: 5′-AGTTGTCACCGAACGAATCTGT-3′, Reverse primer: 5′-CAAGAAGGGGCAGGTACTGAA-3′), followed by enzymatic digestion with *NlaIII* that recognizes the 5′-CATG-3′ in the WT allele (Appendix A). The founder (F0) mosaic embryos were raised to adulthood and outcrossed with WT fish in order to screen for F1 mutants. A single F1 heterozygous fish, which carried a 1 bp deletion mutation in the *fbxl3a* target site (Appendix A), was selected and outcrossed with WT fish. To decrease the risk for off-target mutations, heterozygous F2 and subsequently F3 fish were outcrossed with WT fish.

### 4.5. Establishment of Cell Lines and Bioluminescence Assay

Cell lines were established using enzymatically dissociated embryos obtained from *fbxl3a*^−/−^ as well as WT sibling crosses. Fertilized eggs were washed with sterile E3 medium (5 mM NaCl, 0.17 mM KCl, 0.33 mM CaCl_2_, 0.33 mM MgS0_4_) in the presence of 10^−5^% Methylene Blue. At 24 hpf, embryos were bleached, separated from their chorions and washed with phosphate-buffered saline (PBS). Embryos were then trypsinized (Gibco BRL, Gaithersburg, MD, USA) for 5 min and dissociated tissues were plated in a cell culture flask (Greiner Bio-One GmbH, Kremsmuenster, Austria) in L-15 (Leibovitz) culture medium (Gibco BRL, Gaithersburg, MD, USA) supplemented with 20% serum (Sigma-Aldrich Chemie GmbH, Steinheim, Germany), 2% Penicillin/Streptomycin and 0.2% fungicide (Gentamicin, Gibco BRL, Gaithersburg, MD, USA, 50 mg/mL stock).

Established cell lines were then propagated at 26 °C in L-15 (Leibovitz) medium (Gibco BRL, Gaithersburg, MD, USA) supplemented with 15% Fetal Calf Serum (Sigma-Aldrich Chemie GmbH, Steinheim, Germany). The medium was supplemented with 100 units/mL penicillin, 100 µg/mL streptomycin and 50 µg/mL gentamicin (Gibco BRL, Gaithersburg, MD, USA) in an atmospheric CO_2_, non-humidified cell culture incubator. For cell maintenance, confluent cultures were routinely split after trypsinization with 0.25% (*w*/*v*) Trypsin.

Cell transfections were performed using FuGene HD (Promega, Madison, WI, USA) reagents according to the manufacturer’s protocols. Real-time luciferase assays were performed and analyzed as previously described [40]. In vivo bioluminescence assays were performed using a Topcount NXT automatic scintillation counter (Perkin Elmer Cetus, Norwalk, CT, USA). Data were imported into Microsoft Excel using the ‘‘Import and Analysis’’ macro (S. Kay, Scripps Research Institute).

### 4.6. Analysis of Bioluminescence

Analysis of bioluminescence data was preceded by standardization of bioluminescence of each individual well by subtracting the mean and dividing by the standard deviation, followed by long-term trend removal and smoothing. Period was estimated as the mean time difference between each pair of consecutive peaks of the smoothed bioluminescence function. Phase was estimated as the mean direction of peak time relative to the estimated period [41]. The reported amplitude was defined as half the difference in normalized activity between a selected peak and the preceding trough. Statistical differences in period and amplitude between groups were determined by Wilcoxon rank-sum test, and statistical differences in phase were determined by Watson–Williams test for the homogeneity of means.

The difference in conditions and characteristics between experiments required separate considerations for each experiment as follows. For the *per1b:luc*, activity in mutant *fbxl3a*^−/−^ cells transfected with the exogenous zebrafish FBXL3a expression construct (Figure 5A), both long term trend removal and smoothing were performed via moving average (with 29 and 9 h windows, respectively). A linear mixed model for period was fit, with individual larvae as a random effect and the following three fixed effects: genotype, LD/DD condition and their interaction. Statistical differences were assessed by ANOVA, followed by pairwise comparisons adjusted by the BH procedure [42], controlling FDR at level 0.05. For the *E-box:luc* activity (Figure 5E), both long term trend removal and smoothing were performed via LOESS (with 44- and 22 h windows, respectively). The last maximum point to be considered as LD was the first to follow the 108 h mark (last light period). 

### 4.7. Cloning and Transfection of Zebrafish fbxl3a Coding Sequence

The ORF of zebrafish *fbxl3a* was PCR amplified using the following primers: Forward: 5′-cggttgacattcggttgatt-3′; Reverse: 5′-attaggagtgtcccgtgctg-3′, cloned into a prokaryotic expression vector pJET1.2 (Thermo Fisher Scientific, Waltham, MA, USA) and then subcloned into the *BamHI*-*XhoI* sites of the expression vector HA-pcDNA3.1/myc-HisA (modified pcDNA3.1/myc-HisA, Promega, Madison, WI, USA). Specifically, a 5′HA-tag-*fbxl3a* construct was generated by PCR amplification using the pJET1.2-FBXL3a construct as template and Pfu polymerase (Promega, Madison, WI, USA), with the forward primer creating a *BamHI* site and the ATG of the *fbxl3a* gene mutated to TTG to maintain the protein in frame with the 5′HA-tag (5′-TCCCTCAgGATCCCCAAAATTG-3′) and a reverse T7 primer (5′-TAATACGACTCACTATAGGG-3′) present in the pJET1.2-FBXL3a construct, downstream of the original *fbxl3a* stop codon and the *XhoI* site. PCR was performed with the following parameters: 95 °C for 2 min, followed by 30X (95 °C for 1 min/51.5 °C for 30 s/72 °C for 3 min) and 72 °C for 5 min. The resulting construct was then transfected into zebrafish cells, and expression of the HA-FBXL3a protein (circa 50 kDa) was validated in a Western blot analysis using an anti-HA high affinity rat IgG_1_ antibody (Roche Diagnostics, Penzberg, Germany, Appendix A) according to a previously described protocol [43].

### 4.8. Rhythmic Locomotor Activity Measurements in Zebrafish

Locomotor activity of zebrafish larvae was monitored using the DanioVision observation chamber (Noldus Information Technology, Wageningen, The Netherlands). On the 4th dpf, larvae were placed individually in a 48 well-plate and kept under 12:12 h light/dim light (Ldim) cycles for two days, then exposed to different lighting conditions. During the whole assay, larvae were kept under a constant temperature of 28 °C. The locomotor activity data were generated using EthoVision 13.0 software (Noldus Information Technology, Wageningen, The Netherlands) as the distance moved in cm per 10 min time bins. To determine the extent of fitness of the data to a circadian rhythm, the ratio of the power of the frequency that corresponds to the 24 h period to the sum of powers of all frequencies (G-factor) [18] was determined for each individual larva. A higher G-factor score represents higher confidence that the individual larva exhibits a circadian locomotor activity pattern. Statistical differences in the G-factor distribution between groups were analyzed using Wilcoxon test. Locomotor activity data was normalized by dividing the activity by its mean, and smoothed by a LOESS-smoothed 75th-percentile function [44] with half window widths of 3.33 h (20 sliding points) for the moving percentiles and 8.33 h (50 sliding points) for the LOESS curve. Peaks (local maxima) and troughs (local minima) in the normalized smoothed curves were used to compute the periods, phases and amplitudes. Period was estimated as the weighted mean time difference (in hours) between each pair of consecutive peaks. Phase was estimated as the weighted mean direction (mean of circular quantities [41]) of peak time relative to the estimated period. Since higher amplitudes are less sensitive to noise, weight was assigned to each peak in proportion to its amplitude for estimating period and phase. The reported amplitude was defined as half the difference in normalized activity between the peak of the 2^nd^ day of tracking and the preceding trough. Statistical differences in period and amplitude between groups were determined by Wilcoxon rank sum test, and statistical differences in phase were determined by Watson–Williams test for the homogeneity of means.

### 4.9. Sleep Measurements in Zebrafish

Sleep analysis was performed as previously described [20,22,45]. Sleep value refers to the number of minutes each individual larva spent without moving per one hour, with stop velocity threshold of 0.59 cm/second and start velocity threshold of 0.60 cm/s. Repeated-measures ANOVA was applied to compare sleep time using a significance level of 0.05.

### 4.10. Zebrafish Morphological Examination

The morphology of the fish was characterized using measurements of gross morphology and traits suspected to differ between the WT and *fbxl3a* mutants. Offspring of a cross between a pair of *fbxl3a* heterozygous (*fbxl3*^+/−^) fish were examined under a dissecting microscope at 24 and 48 hpf. The following were identified at this stage: 8 WT, 23 heterozygous and 13 homozygous individual embryos. The remaining embryos where then reared together under the same conditions. At 10 weeks of age, fish were anesthetized in Tricaine and fixed in >90% ethanol. A tissue sample was taken from the caudal or pectoral fin for genotyping by PCR. 6 WT and 9 *fbxl3a* mutants were examined from lateral, dorsal, and ventral aspects, and then cleared and double-stained to visualize skeleton structures [46,47] from the lateral, dorsal and ventral aspects. A total of 22 morphological traits were measured, including body dimensions, head morphology and size of several cranial and body bones (Appendix A), and analyzed using ANOVA.

### 4.11. Whole Mount in Situ Hybridization

Whole mount ISH and quantification of *aanat2* expression was performed as previously described [28,29]. Briefly, embryos (*n* = 10–20 for each time point) were fixed in 4% paraformaldehyde and kept overnight at 4 °C, washed in PBTw and stored in 100% Methanol. Digoxigenin-labeled sense and anti-sense probes were generated for *aanat2* and applied at a concentration of 1 ng/1 μL hybridization buffer overnight at 65 °C. In order to detect ISH signal, stained embryos were placed in 70% glycerol and observed using an Olympus microscope (SZX2) and documented using an Olympus digital camera (DP74). The staining signal was quantified using ImageJ software (National Institute of Health, Bethesda, MD, USA).

### 4.12. Fish and Embryos

Adult zebrafish were raised and maintained in a water system of 28 °C under a 12:12 h LD cycle and fed twice a day. Embryos were produced by placing males and females in a breeding tank in the evening prior to the spawning, and spawning occurred the following morning when lights were turned on. Embryos were collected and kept in a Petri dish with “embryo water” containing methylene blue (0.3 ppm) and stored in a light-controlled incubator at 28 °C. For ISH experiments, PTU (0.2 mM) was applied to prevent pigmentation.

## Figures and Tables

**Figure 1 ijms-23-02373-f001:**
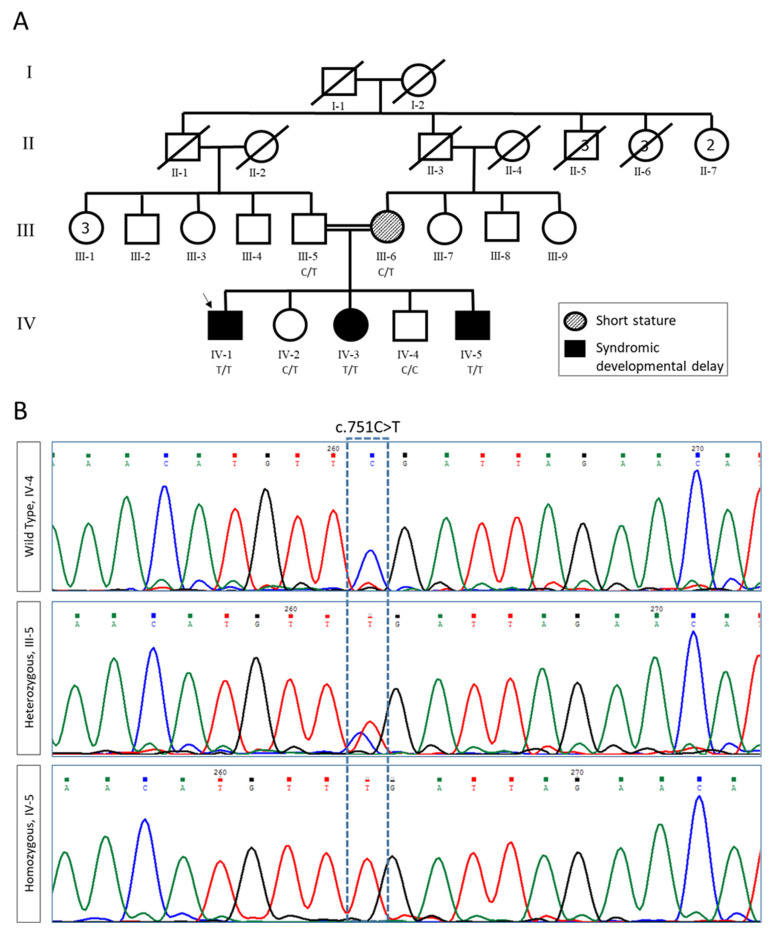
Pedigree and segregation analysis of *FBXL3* variants. (**A**) Pedigree of the studied family. Proband is indicated by an arrow. (**B**) Sanger sequencing of wildtype (WT, IV-4), heterozygous carrier (III-5), and homozygous variant c.751C >T (IV-5).

**Figure 2 ijms-23-02373-f002:**
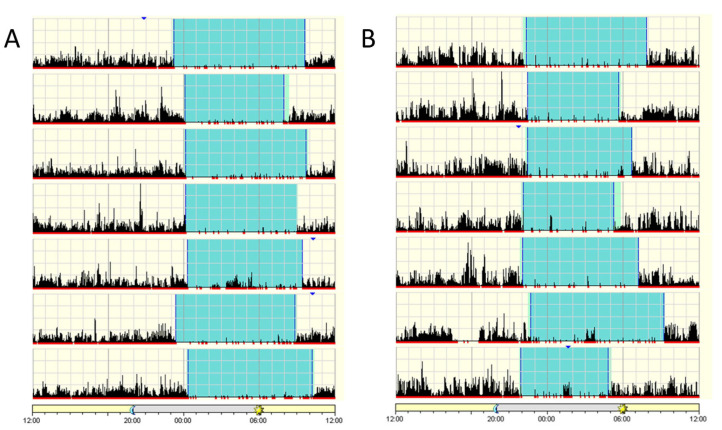
Patient actigraphs. Each horizontal segment in the actigraph represents a 24 h period from noon to noon the next day. The black lines indicate physical activity, with height of the lines corresponding to the occurrence and degree of physical activity. Periods of rest are highlighted in light green and sleep in cyan. (**A**) Patient IV-5, 10-year-old male. (**B**) Patient IV-3, 18-year-old female. Actigraphs show a regular sleep pattern with a set sleep time and mostly minor variations in wake-up time.

**Figure 3 ijms-23-02373-f003:**
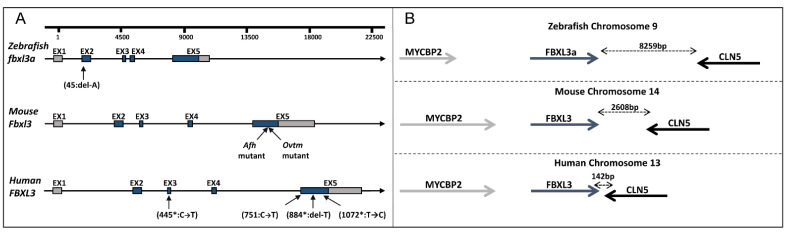
*FBXL3* genomic region in humans, mice and zebrafish. (**A**) Schematic representation of *FBXL3* gene arrangement. Untranslated and coding regions are depicted as grey and blue boxes, respectively; introns are depicted as lines. The position of reported mutations in zebrafish (this study), mouse (*Afh* and *Ovtm* [6,8]) and humans (*Ansar et al. [10], and this study) are indicated. (**B**) Illustration of *FBXL3* conserved synteny with adjacent genes. Bidirectional arrows indicate the physical distance between nearby genes.

**Figure 4 ijms-23-02373-f004:**
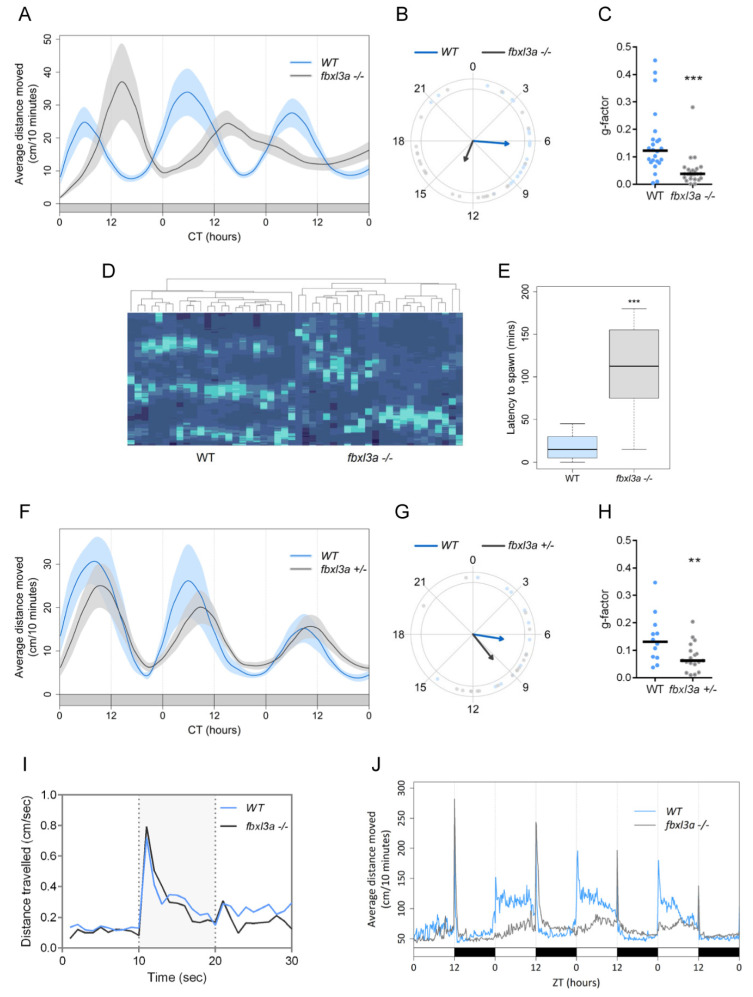
FBXL3a deficiency affects locomotor activity rhythms in zebrafish larvae under different lighting conditions. (**A**) Locomotor activity analysis of 6–8 days post-fertilization (dpf) WT and *fbxl3a*^−/−^ larvae under constant dim light (DimDim) after entrainment to 3 light/dark (LD) cycles, followed by 2 light/dim light (LDim) cycles. The average distance travelled (cm/10 min) is plotted on the y-axis; circadian time (CT, hours) is plotted on the x-axis. Error bars represent ± SE (*n* = 24). (**B**) Phase difference expressed as activity peak with respect to CT0, indicating a 7.2 h phase-delay for the *fbxl3a* mutants (*p* < 0.001, Watson–Williams test). Arrows represent the mean value of each genotype; dots represent individual values; and arrow length is inversely related to the variation within genotype. (**C**) G-factor distribution of WT and *fbxl3a*^−/−^ larvae under DimDim. Each dot represents the G-factor score of an individual larva; the median value for each genotype is indicated by a black horizontal line. A significant difference in G-factor distribution was determined between the two genotypes (*** *p* < 0.001, Wilcoxon test). (**D**) Clustering analysis represented by a heat map emphasizes the disordered rhythms of locomotor activity of *fbxl3a*^−/−^ larvae. Right cluster, *fbxl3a*^−/−^ larvae (*n* = 24); left cluster, WT siblings (*n* = 24). Significant clustering was determined between the two genotypes using Wilcoxon test (*p* < 0.001). (**E**) Delayed spawning time of adult *fbxl3a*^−/−^ fish under LD conditions compared with their WT siblings (*** *p* < 0.001, Wilcoxon test, *n* = 16). (**F**) Locomotor activity analysis of 6–8 dpf WT (*n* = 12) and heterozygous (*fbxl3a^+/^*^−^, *n* = 18) larvae under DimDim after entrainment to three LD cycles, followed by 2 LDim cycles. The average distance travelled (cm/10 min) is plotted on the y-axis; CT (hours) is plotted on the x-axis. Error bars represent ± SE. (**G**) Phase difference expressed as activity peak with respect to CT0, indicating a 2.83 h difference between genotypes. Arrows represent the mean value of each genotype; dots represent individual values; and arrow length inversely represents variation within genotype. (**H**) G-factor distribution of WT and *fbxl3a^+/^*^−^ larvae under DimDim. Each dot represents the G-factor score of an individual larva; the median value for each genotype is depicted as a black horizontal line. A significant difference in G-factor distribution was determined between the two genotypes (** *p* < 0.01, Wilcoxon test). (**I**) Larval mobility is not affected by *fbxl3a* LOF. On the 6th dpf, larvae (*n* = 24) were placed in the DanioVision tracking system. After acclimation to 2 h of light, three 10 s dark flashes, separated by 30 min of light, were applied. The activity of the larvae before, during (grey-shaded) and after the dark flashes was analyzed. No difference was observed between two genotypes, indicating that LOF of *fbxl3a* does not affect larval mobility. (**J**) Blunted locomotor activity rhythms of *fbxl3a*^−/−^ larvae under LD cycles and their lack of response to dark-to-light transitions. The average distance travelled (cm/10 min) is plotted on the y-axis; zeitgeber time (ZT, hours) is plotted on the x-axis. Error bars represent ± SE (*n* = 24). White and black bars indicate light and dark conditions, respectively.

**Figure 5 ijms-23-02373-f005:**
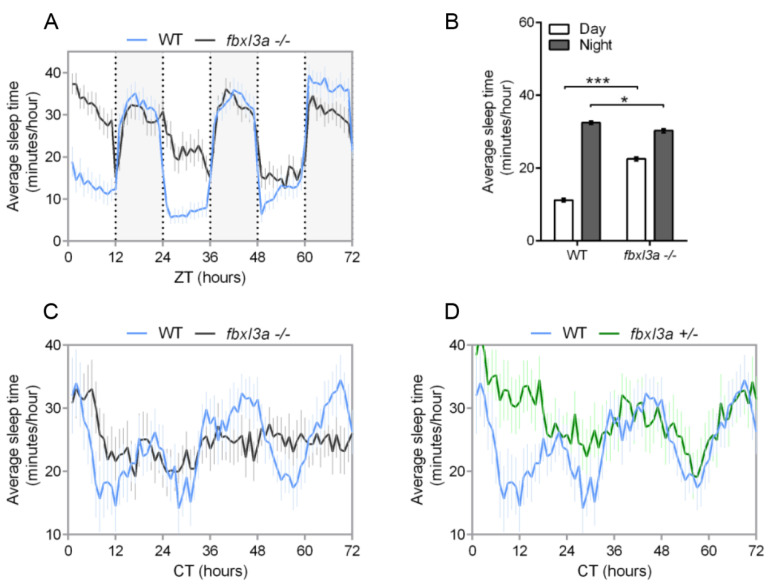
FBXL3a deficiency affects sleep–wake cycles in zebrafish larvae. Sleep analysis of 6–8 dpf *fbxl3a*^−/−^, *fbxl3a**^+/^*^−^ and WT sibling larvae under various lighting conditions. (**A**) Rhythms of sleep under LD are impaired in *fbxl3a*^−/−^ larvae. The average sleep time under LD (minutes/hour) is plotted on the y-axis and ZT (hours) is plotted on the x-axis. Error bars represent ± SE (*n* = 24). White and grey-shaded areas indicate light and dark, respectively. (**B**) The average sleep time (±SE, *n* = 24) for total daytime and total nighttime for each genotype under LD conditions. Significant differences in day/night sleep time alterations were found between *fbxl3a*^−/−^ larvae and their WT siblings (*** *p* < 0.001 and * *p* < 0.05 for day and night, respectively, repeated-measures ANOVA). (**C**) Circadian rhythms of sleep under DimDim, after entrainment by 3 LD and 2 LDim cycles, are impaired in *fbxl3a*^−/−^ larvae (*n* = 18) compared to WT siblings (*n* = 12). The average sleep time (minutes/hour) under DimDim is plotted on the y-axis and CT (hours) is plotted on the x-axis. Error bars indicate ± SE. (**D**) Circadian rhythms of sleep under DimDim are impaired in *fbxl3a**^+/^*^−^ larvae (*n* = 18) compared to WT siblings (*n* = 12), the WT trace is the same as in C. Sleep measurement was conducted after entrainment by 3 LD and 2 LDim cycles. The average sleep time (minutes/hour) under DimDim is plotted on the y-axis and CT (hours) is plotted on the x-axis. Error bars represent ± SE.

**Figure 6 ijms-23-02373-f006:**
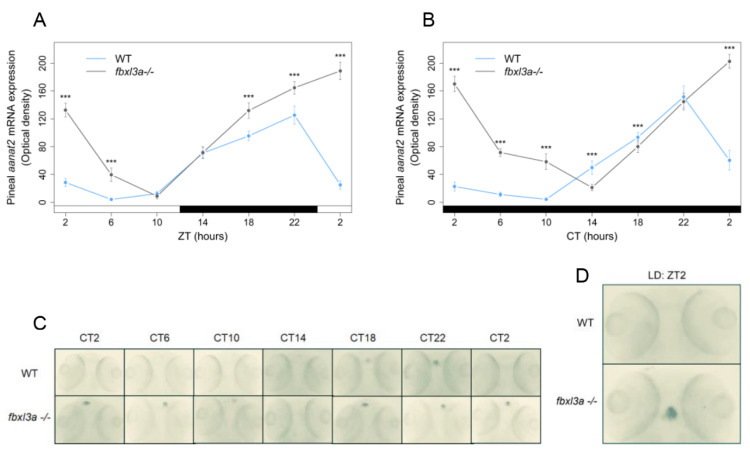
Clock-controlled *aanat2* expression in the zebrafish pineal gland is affected by FBXL3a. *Fbxl3a* mutant larvae exhibit abnormal pineal *aanat2* mRNA expression under constant dark (DD) and LD photoperiodic regimes (*** *p* < 0.001, *t*-test). Each value represents the mean optical density of pineal *aanat2* expression using whole mount in situ hybridization (ISH) quantification (*n* = 15–20). ZT (hours) is indicated for each sample; white and black bars represent light and dark, respectively. Pineal *aanat2* ISH quantification of 72–96 h post-fertilization (hpf) *fbxl3a*^−/−^ larvae and their WT siblings under LD (**A**) or DD (**B**) conditions, after entrainment by three LD cycles. (**C**) Representative pineal *aanat2* ISH signal of 72–96 hpf *fbxl3a*^−/−^ larvae (bottom panel) and their WT siblings (top panel) under DD conditions. (**D**) Pineal *aanat2* signal of a 96 hpf *fbxl3a*^−/−^ larva and its WT sibling after four LD cycles (at ZT2).

**Figure 7 ijms-23-02373-f007:**
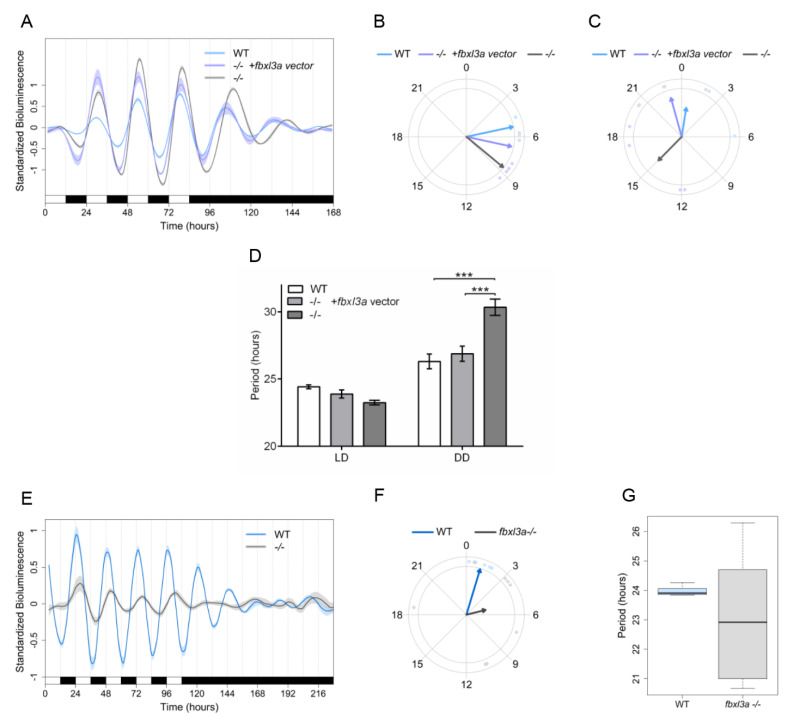
FBXL3a LOF disrupts core molecular oscillation in zebrafish cells. (**A**) Impaired *per1b* promoter activity in *fbxl3a*^−/−^ cell line is partially rescued by exogenous FBXL3a. Standardized luciferase activity in *per1b:luc*-transfected cell lines derived from *fbxl3a*^−/−^ and their WT siblings under different lighting conditions. Cells were transfected with the *per1b:luc* construct and with a FBXL3a expression vector or empty vector. Error bars represent ±SE (*n* = 4). (**B**) Phase difference under LD conditions is expressed as activity peak with respect to ZT0, indicating a 3.45 h phase-delay for the *fbxl3a*^−/−^ cell line, which is partially rescued by exogenously expressed FBXL3a, yielding a 1.61 h phase-delay. Arrows represent the mean value of the studied cell line, dots represent individual well values, and arrow length inversely represents the variation among wells of the same cell line. (**C**) Phase difference under DD conditions is expressed as activity peak with respect to CT0, indicating a 14.38 h phase-delay of the *fbxl3a*^−/−^ cell line, which is partially rescued by exogenous FBXL3a, inducing a 1.63 h phase advance. Arrows represent the mean value of the studied cell line, dots represent individual well values, and arrow length inversely represents the variation among wells of the same cell line. (**D**) Mean period values of bioluminescence rhythms of cells transfected with *per1b:luc* construct and with the FBXL3a expression vector or empty vector. Period (hours) under LD: WT = 24.42, *fbxl3a*^−/−^ = 23.24, *fbxl3a*^−/−^ + FBXL3a expression vector = 23.88; Period (hours) under DD: WT = 26.31, *fbxl3a*^−/−^ = 30.34, and *fbxl3a*^−/−^ + FBXL3a expression vector = 26.88. Error bars represent ± SE (*n* = 8). Statistical differences were assessed by ANOVA, followed by pairwise comparisons adjusted by the BH procedure controlling FDR at level 0.05 (*** *p* < 0.001). (**E**) Standardized bioluminescence in *E-box:luc*-transfected cell lines derived from *fbxl3a*^−/−^ and their WT siblings under five LD cycles followed by 5 days of DD. Error bars represent ± SE (*n* = 8 wells). (**F**) A delayed phase of the synthetic E-box promoter activity in the *fbxl3a*^−/−^ cell line under LD. Phase difference expressed as activity peak with respect to ZT0, indicating a 4.07 h difference between cell line genotypes. Arrows represent the mean value of the studied cell line, dots represent individual well values, and arrow length inversely represents the variation among wells of the same cell line. Significance was determined using Watson–Williams test (*p* < 0.001). (**G**) Abnormal period distribution of *E-box:luc* expression in *fbxl3a*^−/−^ cell line. Mean period values and distribution of luciferase activity under five LD cycles (WT = 23.91 IQR = 0.38, *fbxl3a* mutant = 23.37 IQR = 7.29).

## Data Availability

The human *FBXL3* variant presented in this study is available from the ClinVar database [accession number VCV001177304.1].

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
