# Peer review of "A Zebrafish Model for a Rare Genetic Disease Reveals a Conserved Role for FBXL3 in the Circadian Clock System"

_ijms, 2022, doi:10.3390/ijms23042373_

Round 1

Reviewer 1 Report

Despite its length, the article is written objectively and clearly, making a good description of the state of the art and justifying the objective of the work. The results are also clearly described, and the figure legends are complete and independent. The findings pointing to an evolutionary conserved role for FBXL3 in the circadian clock system across vertebrates and to the acquisition of developmental roles in humans are interesting. However, I only have two suggestions: in the Introduction section please indicate what BMAL, CLOCK, PER and CRY stands for; and in the end of the Discussion please add a small conclusion paragraph or at least make it more evident.

Author Response

We are very happy to read the positive comments of Reviewer 1 regarding the content and conclusions of our manuscript. We have completely addressed the specific suggestions of this reviewer:

“I only have two suggestions: in the Introduction section please indicate what BMAL, CLOCK, PER and CRY stands for; and in the end of the Discussion please add a small conclusion paragraph or at least make it more evident.”

We have adjusted the text in the following way:

  1. In the Introduction text, we now spell out the complete names for the acronyms of the BMAL, CLOCK, PER and CRY clock genes.
  1. At the end of the discussion section, we have now included the following short Conclusion statement: In conclusion, the phenotype of LOF FBXL3 mutations in zebrafish reinforce the notion that in vertebrates, FBXL3 plays a highly conserved role in maintaining normal circadian clock oscillations. However, the morphological and neurodevelopmental affects observed in the human patients point to some degree of species-specific diversity in FBXL3 function that extends beyond the circadian clock. (Lines 447–451).

Reviewer 2 Report

In this study, authors investigated the effect of FBXL3 LOF mutation in zebrafish in comparison with other findings in humans and mice. The introduction does not clearly justify the importance of the study and the objective is not very clear. The methods are well described, except for the Exome analysis section, which is poorly described as it cannot offer sufficient information for the reader to understand how the sequencing and analysis of the sequence data were performed. Results and discussion were well written and addressed accordingly. For this reason, I recommend the manuscript for publication with small improvements in the introduction section, and if possible in the section Exome analysis in the Material and methods.

For the introduction I specifically recommend the following:

Page 4, Lines 121-128 must be merged to the page 2, lines 66-70 in the introduction section, and modified accordingly to avoid redundancy. It has a clear aim and justification for the study and introductory style that should not be in the results section. Therefore, the paragraph starting in page 4, line 121 should start in the line 128 with the sentence: “Zebrafish fbxl3a (RefSeq NM_001005773.3) encodes…”.

Author Response

We are grateful for the very constructive advice of this reviewer:

“The introduction does not clearly justify the importance of the study and the objective is not very clear. The methods are well described, except for the Exome analysis section, which is poorly described as it cannot offer sufficient information for the reader to understand how the sequencing and analysis of the sequence data were performed.”

and

“For the introduction I specifically recommend the following:

Page 4, Lines 121-128 must be merged to the page 2, lines 66-70 in the introduction section, and modified accordingly to avoid redundancy. It has a clear aim and justification for the study and introductory style that should not be in the results section. Therefore, the paragraph starting in page 4, line 121 should start in the line 128 with the sentence: “Zebrafish fbxl3a (RefSeq NM_001005773.3) encodes...”.

We have adapted our text precisely as they suggested.

  1. The Introduction text has been modified precisely as recommended by the reviewer. Specifically, the final section of the Introduction now reads:

“Given the reported contribution of FBXL3 function to the molecular clock in mice and in human cell lines, we also monitored their sleep pattern but found no evidence for any sleep disturbances. Therefore, in humans the most obvious phenotypes of the FBXL3 mutation are developmental delay and morphological abnormalities, while in the mouse model only disrupted circadian rhythms have been reported [6,8]. However, in both mice and human cell lines, the role of FBXL3 is to regulate the pace of CRY turnover and of the molecular oscillator [7,8,11].

In order to scrutinise the effect of FBXL3 LOF on both morphogenesis and the circadian clock in more detail, we next chose to investigate the role of FBXL3 in zebrafish. In addition to being a powerful model for studying developmental genetics and morphological abnormalities [12-14], the zebrafish is a diurnal species that has been widely used as a model for circadian clock research [15-17].” (Lines 65–79).

As recommended, this involved moving the text that originally started the results section 2.3. (Establishing a FBXL3-deficient zebrafish line) to the Introduction, so this short text block has now been deleted from section 2.3 and this section now begins with Zebrafish fbxl3a (RefSeq NM_001005773.3) encodes a 431 amino acid……” (lines 140–152).

  1. The exome sequence analysis section in the Materials and Methods has now been completely rewritten as follows:

Following informed consent, exome sequencing analysis was performed on DNA extracted from whole blood of the proband (individual IV-1) and the parents, as previously described [37-39]. Exonic sequences were enriched from the genomic DNA samples using SureSelect Human All Exon v.5 50 Mb Kit (Agilent Technologies, Santa Clara, CA). Sequences were determined by HiSeq2500 sequencing system (Illumina, San Diego, CA) as 100 bp paired‐end runs. Data analysis including read alignment and variant calling was performed by DNAnexus software (Palo Alto, CA) using the default parameters with the human genome assembly hg19 (GRCh37) as reference. Filtering was performed as described [37]. The trio exome analysis yielded 61.0 million mapped reads, with a mean coverage of 81X.” (Lines 454–463).

Reviewer 3 Report

Authors have identified a FBXL3 mutation in patients with syndromic developmental delay and found that they show normal sleep pattern. This study produced fbxl3a mutant zebrafish and analyzed them at molecular, cellular, and behavior (sleep) levels. The mutant fish have defects in circadian clock system, whereas they did not show a defect in development. Authors discussed the difference in phenotypes between patients with a FBXL3 mutation and fbxl3a mutant zebrafish. The experiments are well-organized and the conclusion is strongly supported by the results. This reviewer recommends the manuscript for publication. However, I would like authors to address minor comments described below.

1) In figures 7A-7D, the efficiency of rescuing the bioluminescence phase-delay and period lengthening in the fbxl3a-deficient cells by exogenous expression of FBXL3 is low. I would like authors to discuss reason for this. In addition, confirmation of expression of exogenous FBXL3 would be required.

2) Detailed explanation of background authors focused on patients with syndromic developmental delay would be helpful for readers to understand importance of current study.

Author Response

We are grateful for this Reviewer's constructive advice. They raised the following points:

“1) In figures 7A-7D, the efficiency of rescuing the bioluminescence phase-delay and period lengthening in the fbxl3a-deficient cells by exogenous expression of FBXL3 is low. I would like authors to discuss reason for this. In addition, confirmation of expression of exogenous FBXL3 would be required.

2) Detailed explanation of background authors focused on patients with syndromic developmental delay would be helpful for readers to understand importance of current study.”

We have now completely addressed the issues raised:

  1. Related to our cell line data presented in Figure 7, we have now added some text to the discussion to mention the low efficiency of rescue in the transient transfection assays and to speculate on possible explanations:

“Importantly, this disruption was partially rescued by ectopic expression of WT FBXL3a protein. The relatively low efficiency of rescue is most likely the consequence of the transient transfection of the cells with the FBXL3a expression construct which typically results in heterogeneity of expression levels within individual transfected cells. Nevertheless, the effect of the fbxl3a mutation on the molecular oscillator in zebrafish cells is consistent with similar affects in human- and mouse-derived cell lines and points to a highly conserved contribution of FBXL3 to clock function in vertebrates. (Lines 374–381).

In addition, we have now included an addition supplementary Figure (S3) which presents our data from western blotting analysis of protein extracts from the transiently transfected cells (and appropriate controls) using an anti-HA epitope tag antibody that clearly shows the ectopically expressed FBXL3 protein. The reviewer was absolutely right to point out the need for this important control experiment.

  1. The reasons for unravelling the genetic cause of this rare genetic disease are now explained in lines 90–92:

“With the goal of understanding the etiology of this rare neurodevelopmental syndrome, and to enable future identification of carriers, we chose to study the genetic basis of this disease.”